# Volatilization of Dicamba Diglycolamine Salt in Combination with Glyphosate Formulations and Volatility Reducers in Brazil

Caio Antonio Carbonari [1,*], Renato Nunes Costa [2], Bruno Flaibam Giovanelli [2], Natalia Cunha Bevilaqua [2], Matheus Palhano [3], Henrique Barbosa [4], Ramiro Fernando Lopez Ovejero [3] and Edivaldo Domingues Velini [1]

1 Department of Crop Science, College of Agricultural Sciences, São Paulo State University (Universidade Estadual Paulista "Júlio de Mesquita Filho" UNESP), Botucatu 18610-034, SP, Brazil; edivaldo.velini@unesp.br
2 College of Agricultural Sciences, São Paulo State University (Universidade Estadual Paulista "Júlio de Mesquita Filho" UNESP), Botucatu 18610-034, SP, Brazil; renatonunes12@hotmail.com (R.N.C.); bfgiovanelli@yahoo.com.br (B.F.G.); nataliacunha_8@hotmail.com (N.C.B.)
3 Bayer CropScience, São Paulo 04761-000, SP, Brazil; matheus.palhano@bayer.com (M.P.); ramiro.ovejero@bayer.com (R.F.L.O.)
4 Bayer CropScience, São José dos Campos 12241-421, SP, Brazil; henrique.barbosa@bayer.com
* Correspondence: caio.carbonari@unesp.br; Tel.: +55-1438807223

**Abstract:** Dicamba can be included in weed management programs for Brazilian agricultural crops, such as *Conyza* spp. and *Amaranthus* spp., and it is essential to implement good management practices that include salts with low volatility levels and appropriate associations to maintain volatility at acceptable levels. The objective of this study was to evaluate the volatilization of dicamba diglycolamine (DGA) salt associated with different glyphosate salts and volatility reducers. Laboratory and field studies were conducted with the application of DGA alone and in mixtures with three glyphosate formulations (potassium salt, ammonium salt, and di-ammonium salt) with and without a volatility reducer. Under laboratory conditions, the sprayed targets (corn straw) were sent to a vapor collection system for subsequent determination of the amount of volatilized dicamba. In the field, the treatments were applied in a tray containing clay soil, and then, these trays were arranged in soybean rows for 48 h under plastic tunnels. The plant injury and the concentrations of the dicamba in the soybean plants at different distances from the tray were determined. The methodologies used in this study were adequate for understanding the volatility of DGA. The volatility of the dicamba DGA salt used was reduced and was managed through the use of volatility reducers and the correct formulation of glyphosate in the mixture. The VR was efficient in reducing the volatility for dicamba alone and DGA in combination with all glyphosate salts. The combination of dicamba DGA salt with glyphosate potassium salt and a volatility reducer was the mixture with the lowest volatility and is the most suitable combination to recommend to farmers.

**Keywords:** adjuvants; auxin herbicide; formulation; soybean; vapor

## 1. Introduction

Dicamba (3,6-dichloro-2-methoxybenzoic acid or 2-methoxy-3,6-dichlorobenzoic acid) is a systemic herbicide in the group of synthetic auxins used to control annual and perennial broadleaf weeds in the postemergence of weeds. It can be used during preplanting of soybean and cotton crops [1] not tolerant to dicamba, as long as the recommendations for the dose and safety interval defined on the package insert of the products are followed. In recent years, with the launch of Xtend® technology, that allows the application of dicamba in postemergence soybean and cotton crops, there has been an intensification in the use of this herbicide in the United States of America. In Brazil, the Ministry of Science, Technology, Innovation, and Communications (CTNBio) deregulated soybean tolerance to dicamba and glyphosate in 2018 [2,3]. With the launch of this crop in Brazil, only the application of dicamba in the preplanting area of the crop was authorized by herbicide registration [4].

Dicamba is a plant auxin-mimicking herbicide that stimulates cell elongation and differentiation, leading to the rapid growth of stems, petioles, and leaves [5,6]. This abnormal growth of the plant disrupts the cellular transport systems and may lead to the death of the plant [7]. Nontarget plants susceptible and exposed to even a small amount of dicamba may experience phytotoxic effects with deformities (cupping effect) and epinastia [8,9].

Dicamba has a low organic carbon partition coefficient (Koc) and, therefore, a low affinity for soil particles and suspended sediments [10] (Table 1). Furthermore, dicamba also has a low octanol/water partition coefficient (Kow) and is resistant to oxidation and hydrolysis under most conditions [11,12]. In its acid form, it is characterized as a moderately volatile compound [6]. The degree of volatility depends on several factors, including the amount applied, atmospheric temperature, atmospheric humidity, chemical formulation, and surface on which it is applied [13].

**Table 1.** Physicochemical characteristics of dicamba (not including its salts).

| Characteristics | |
|---|---|
| Chemical group | Benzoic acid |
| Molecular weight (g mol$^{-1}$) | 221.04 |
| Name IUPAC | 3,6-dichloro-o-anisic acid |
| CAS name | 3,6-dichloro-2-methoxybenzoic acid |
| Solubility in water, 20 °C (mg L$^{-1}$) | 250,000 |
| Melting point (°C) | 115 |
| Boiling point (°C) | Decomposes before |
| Degradation point (°C) | 230 |
| Octanol-water partition coefficient (Kow) at pH 7, 20 °C (Log P) | −1.88 |
| Density (g mL$^{-1}$) | 1484 |
| Dissociation constant—pKa at 25 °C | 1.87 (strong acid) |
| Vapor pressure, 20 °C (mPa) | 1.67 |
| Henry's law Constant at 25 °C (Pa m$^3$ mol$^{-1}$) | $1.0 \times 10^{-04}$ |
| Soil adsorption coefficient (Koc) (L kg$^{-1}$) | 1.42 |
| Chemical formula | $C_8H_6Cl_2O_3$ |
| Chemical structure |  |

Source: PPDB: Pesticide Properties Database: University of Hertfordshire [14], Koc [10].

Dicamba is a strong acid (pKa 1.87) that is formulated as a salt, and the acid form is the volatile form [15–17]; thus, its molecular state can have a substantial impact on its volatility [16,18]. This herbicide is used in agriculture after the neutralization of acid in a soluble salt. Currently, there are some dicamba formulations, including dimethylamine salt (DMA), diglycolamine salt (DGA), and N,N-bis-(3-aminopropyl) methylamine (BAPMA), that differ significantly in terms of their volatility levels. Researchers have demonstrated a significant reduction in the volatility of the DGA and BAPMA when compared to that of the DMA [18,19].

A volatility reducer (VR) has also been developed to further decrease the volatility profile of dicamba, eliminating free protons in the dicamba spraying solution [20]. This technology can be added to the product formulation process or applied as a mixture in the application tank, with the latter being the recommendation for Brazil [4].

Dicamba and other auxinic herbicides, such as 2,4-D, are often applied by farmers in combination with glyphosate in tank mixtures to increase efficacy or broaden the spectrum of weed control. Due to the selection of buva plants (*Conyza sumatrensis*) resistant to 2,4-D in Brazil [21], dicamba has become an important tool in the control of this species as well

as other broadleaf species. Cantu et al. [22] demonstrated that dicamba, in a mixture with glyphosate, showed one of the best results for the control of *C. sumatrensis* during the preplanting of soybean crops.

Under controlled conditions, Mueller and Steckel [18] evaluated the application of (1) DGA dicamba + glyphosate (potassium salt), (2) DGA + glyphosate + VR and (3) DGA + VR in sandy soil at different temperatures. The authors observed that the volatilization of DGA + glyphosate and DGA + glyphosate + VR was higher than that of DGA + VR, indicating that the use of glyphosate in the mixture may increase the volatilization of dicamba.

The combination of dicamba with different glyphosate salts can influence the volatilization process and cause losses to the environment and possible injuries to sensitive crops, especially under more adverse environmental conditions. In the Brazilian market, there are several commercial formulations of glyphosate available in the forms of isopropylamine, di-ammonium, ammonium, potassium, and dimethylamine [23]. In Brazil, the package insert recommends potassium salt glyphosate as the only formulation for use in a tank mixture with dicamba [4]. In the USA, the glyphosate salts dimethylamine, isopropylamine, and ammonia are not recommended to be included in a mixture with dicamba for soybean [24]. The different glyphosate formulations may have different pH values because the solubility of these salts is different and the pH of the glyphosate solutions increases with dilution [25]. Thus, in the context of dicamba volatility, as the pH decreases, hydrogen ions become more readily available and allow the faster formation of dicamba acid, which is prone to volatilization [26]. However, there is no consensus on the direct relationship between the decrease in pH and the increase in dicamba volatility [27].

Nevertheless, with the greater adoption of the no-tillage system in soybean, corn, and cotton and the double-crop system in the Cerrado region (soybean/corn), a significant part of dicamba will be deposited in straw. No-tillage systems in Brazil cover approximately 33 million hectares and continue to be expanded [28], and the interaction of the herbicide with the straw cover surface is different from that on the soil or leaf surface [29].

Some studies in the literature have shown that the addition of glyphosate to dicamba liquid increases its volatilization [17,18]; however, it is not specified whether the difference between the glyphosate formulations available on the market can affect the amount of volatilized dicamba, and the efficiency of the VR in these mixtures has also not been determined. Therefore, the objective of the study was to evaluate the influence of different glyphosate salts (potassium, ammonium, and di-ammonium salts) on the volatilization of DGA, with and without the addition of a volatility reducer.

## 2. Materials and Methods

### 2.1. Studies Conducted in the Laboratory

2.1.1. Application of Treatments

The products were applied using an automated sprayer located in a closed environment, with a metal structure for the support of the 2 m long spray boom. Four XR 11002 VS (TeeJet, Springfield, IL, USA) nozzles were used, with 0.5 m spacing and a 0.5 m height in relation to the targets. The working pressure adopted was 150 kPa, and the forward speed was 3.6 km h$^{-1}$, with a spray volume of 200 L ha$^{-1}$.

The different treatments were applied to corn straw fragments with dimensions of $6 \times 8$ cm. At the time of application, the edges were covered with 1 cm, resulting in a useful area of 35 cm$^2$. The experiment was performed in duplicate in a completely randomized design with eight treatments and three replicates, with each experimental unit consisting of an applied target. The treatments consisted of the application of dicamba (DGA) alone and DGA in a mixture with a VR and glyphosate formulated with different salts (potassium, ammonium, and di-ammonium) (Table 2).

**Table 2.** Different treatments applied to corn straw.

| Treatments | Dose of Herbicide (g a. e. ha$^{-1}$) | Concentration of VR [a] (% *v/v*) | pH of the Solution |
|---|---|---|---|
| DGA [b] (diglycolamine salt) | 720 | - - | 7.06 |
| DGA + VR | 720 | 0.50 | 6.65 |
| DGA + glyphosate (potassium salt) [c] | 720 + 1440 | - - | 4.70 |
| DGA + glyphosate (potassium salt) + VR | 720 + 1440 | 0.50 | 5.24 |
| DGA + glyphosate (ammonium salt) [d] | 720 + 1440 | - - | 3.72 |
| DGA + glyphosate (ammonium salt) + VR | 720 + 1440 | 0.50 | 5.09 |
| DGA + glyphosate (di-ammonium salt) [e] | 720 + 1440 | - - | 6.40 |
| DGA + glyphosate (di-ammonium salt) + VR | 720 + 1440 | 0.50 | 6.48 |

[a] VR: volatility reducer—acetic acid/acetate, VaporGrip® (added in tank mixture) (Monsanto). [b] DGA: digicolamine salt dicamba (Atectra®—BASF). [c] Glyphosate potassium salt (Roundup Transorb R®—Monsanto). [d] Glyphosate ammonium salt (Roundup WG ®—Monsanto). [e] Glyphosate di-ammonium salt (Roundup Original Mais ®—Monsanto).

2.1.2. Vapor Collection System

The steam collection system consisted of a closed system of PVC pipes with inlets to hold 24 cartridges and an outlet containing a capillary (3 mm) connected to a vacuum pump (air flow of 30 mL min$^{-1}$) [29]. This system was placed inside a chromatograph oven that was used as a precision chamber to maintain a constant temperature at 40 °C. The cartridges, where the applied targets were placed, had a length of 19.2 cm and a volume of 132 cm$^3$. The cartridges were closed with a lid containing an opening of 3 mm in diameter to ensure the passage of air through the system. At the opposite end of the cartridge, two filters of PVDF 0.20 μm, 25 mm (Chromafill Xtra, MN, Düren, Germany) were used to sample the dicamba vapor in the air. The filters were arranged in series to ensure high vapor collection efficiency in each cartridge. From the analysis of the two filters in sequence, it was possible to calculate the collection efficiency of the filters [28].

After application, the targets remained at rest for 10 min for drying. At the end of this period, the straw edge protection was removed. Each experimental unit was placed inside a cartridge that was sealed to prevent leakage of the volatilized herbicide. The cartridges containing the targets and the respective filters were placed in the steam collection system and kept in the oven at a constant temperature of 40 °C and relative humidity of 20% for a period of 24 h. After this period of 24 h, dicamba was extracted from the straw, cartridges, and filters.

2.1.3. Extraction of Dicamba from the Targets, Filters, and Cartridges

The straw fragments (target) were removed from the cartridges and placed in a 50 mL centrifuge tube to which 40 mL of methanol: distilled water (25:75 *v/v*) solution was added. The tubes were subjected to an ultrasound bath for 30 min, and then, the solution was transferred to other containers [29]. To determine the amount of dicamba that was retained on the walls of the cartridges, 10 mL of the extraction solution was added to the syringes, which were sealed at both ends for subsequent agitation for 5 min on a shaker table. To determine the volatilized dicamba, the two filters were individually washed using 1.5 mL of the extraction solution, generating samples for filter 1 (first filter of the series) and filter 2 (last filter of the series) that were stored. All extracted solutions were filtered using 0.45 μm Millipore syringe filters and transferred to 2 mL vials.

*2.2. Low-Tunnel Field Trial*

2.2.1. Preparation of the Experimental Area

The experiment was conducted in an experimental area at the Experimental Lageado Farm (UNESP) in the municipality of Botucatu, São Paulo, Brazil (22°50′38.60″ S and 48°25′29.00″ W) at 779 m in altitude. The soil of the experimental area is classified as a dystrophic Red Nitosol (dRN) and has the following physicochemical characteristics: pH (CaCl$_2$) = 4.9; organic matter = 22 g dm$^{-3}$; P = 22 g dm$^{-3}$; Al = 17 g dm$^{-3}$; K = 3.26 g dm$^{-3}$;

Ca = 33 mmol dm$^{-3}$; Mg = 14 mmol dm$^{-3}$; base sum = 50 mmol dm$^{-3}$; cation exchange capacity = 102 mmol dm$^{-3}$; soil base saturation (V%) = 49; sand = 373 g kg$^{-1}$; clay = 469 g kg$^{-1}$, and silt = 158 g kg$^{-1}$.

Soybean (sensitive to dicamba TMG 7026 IPRO) sowing occurred on 16 December 2019, using 16 seeds m$^{-1}$ and a spacing of 0.45 m between rows. Basic fertilization (300 kg ha$^{-1}$, formulation 2:20:20 (N:P:K), pest control, and disease control were performed according to the technical recommendations for soybean cultivation in the region. In the absence of rain, sprinkler irrigation was used to provide an adequate water supply (10 mm, when necessary). The field experiment was conducted in a randomized block design with four replicates.

### 2.2.2. Application of Treatments

The herbicides and mixtures used in this study were the same as those used in the laboratory, but the doses of the herbicides were four times higher (Table 3), aiming at a critical condition for volatilization [30]. The application occurred in trays (60 cm × 30 cm) containing soil from the planting area during soybean stage V6. The soil of the trays was previously saturated with water to field capacity. The spraying of the treatments was performed using an automated sprayer located in a closed environment, with a metal structure to support the 2 m long spray boom. Four XR 11002 VS (TeeJet, Springfield, IL, USA) nozzles were used, with 0.5 m spacing and a 0.5 m height in relation to the targets. The working pressure adopted was 150 kPa, and the forward speed was 3.6 km h$^{-1}$, with a spray amount of 200 L ha$^{-1}$. The application was performed at a 300 m distance and in a closed environment to avoid any contamination of the experimental area.

**Table 3.** Treatments applied to trays with soil.

| Treatments | Dose of Herbicide (g ha$^{-1}$) | Concentration of VR [a] (% *v/v*) | pH of the Solution |
|---|---|---|---|
| DGA [b] (diglycolamine salt) | 2880 [f] | - - | 6.95 |
| DGA + VR | 2880 | 0.50 | 6.76 |
| DGA + glyphosate (potassium salt) [c] | 2880 + 5760 | - - | 4.82 |
| DGA + glyphosate (potassium salt) + VR | 2880 + 5760 | 0.50 | 5.13 |
| DGA + glyphosate (ammonium salt) [d] | 2880 + 5760 | - - | 3.99 |
| DGA + glyphosate (ammonium salt) + VR | 2880 + 5760 | 0.50 | 4.79 |
| DGA + glyphosate (di-ammonium salt) [e] | 2880 + 5760 | - - | 6.37 |
| DGA + glyphosate (di-ammonium salt) + VR | 2880 + 5760 | 0.50 | 6.44 |

[a] VR: volatility reducer—acetic acid/acetate, VaporGrip® (added in tank mixture) (Monsanto). [b] DGA: digicolamine salt dicamba (Atectra®—BASF). [c] Glyphosate potassium salt (Roundup Transorb R®—Monsanto). [d] Glyphosate ammonium salt (Roundup WG®—Monsanto). [e] Glyphosate di-ammonium salt (Roundup Original Mais®—Monsanto). [f] Dose 4 times higher than regular dose for a critical condition for volatilization.

On the same day of application, 1.0 m wide, 6.0 m long, and 1.5 m high plastic tunnels were installed in the field, composed of a PVC tubular structure covered with plastic films. Within the tunnels, the soybean central line was removed, leaving two soybean lines with a distance of 0.9 m between them. This methodology has commonly been used by those in academia and industry to study dicamba volatility [30–32].

The trays were transported to the field immediately after application, and two trays with the respective treatments were installed in the center of each plastic tunnel. The covers of the plastic tunnels were maintained for 48 h and then removed. During this period, thermometers with temperature records were installed outside the plastic tunnels and in the center of the tunnels both 1.2 m aboveground and at the ground level of the trays inside the plastic tunnels, and thermal images were obtained using an infrared camera, Flir T540 (FLIR Systems, Inc., Wilsonville, OR, USA) (Figure 1).

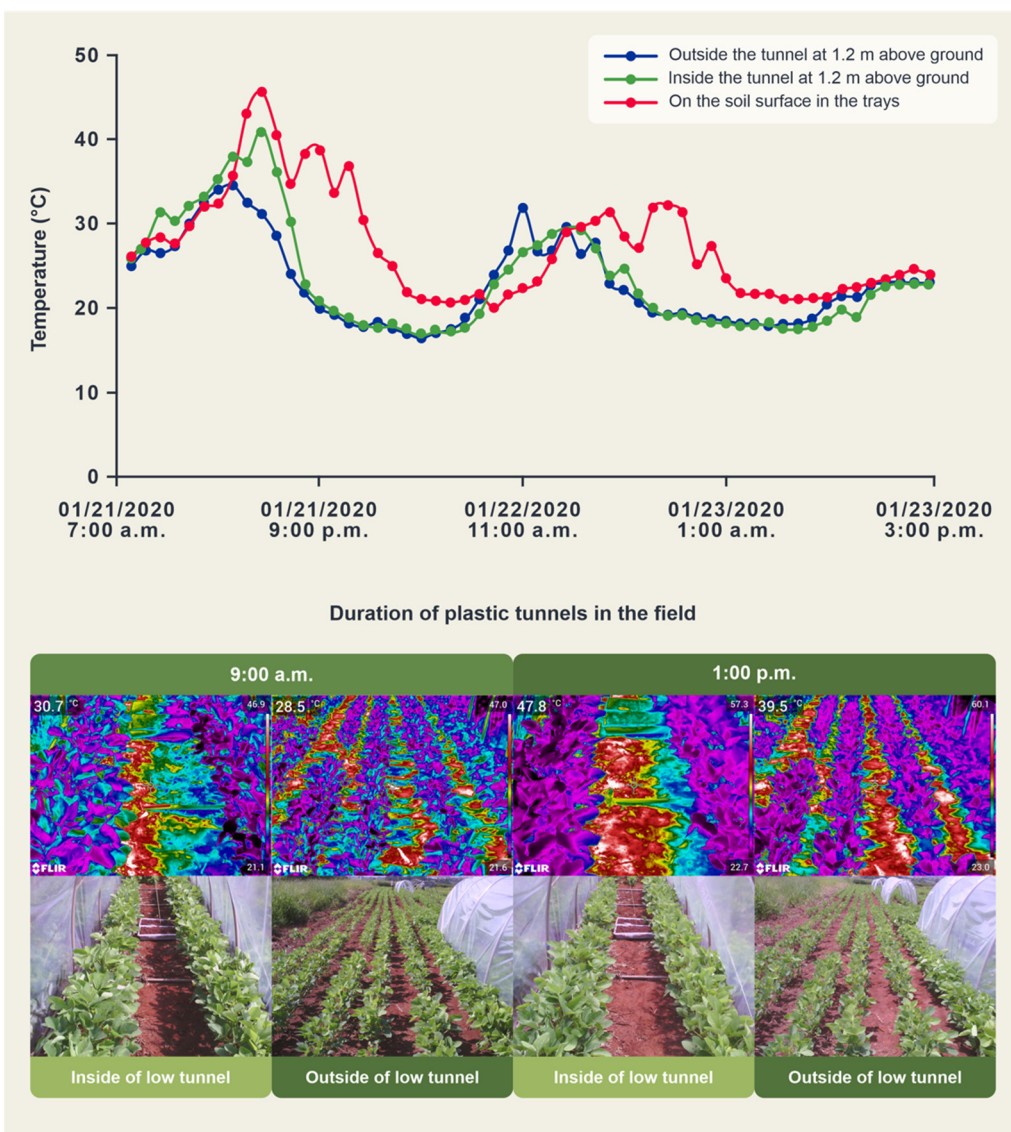

**Figure 1.** Temperatures and thermal images inside and outside the low tunnels.

### 2.2.3. Evaluations

Injury evaluations of soybean plants in both rows were performed at different distances every 0.3 m to 7.5 m from the trays that received the applications. These evaluations were performed at 14 days after application. The visual effects on the plants were classified on a scale from 0 to 100%, with 0 being symptom-free and 100 being a dead plant.

For the dicamba analysis, five fully expanded trifoliate leaves were collected in the upper, middle, and lower thirds at 48 h after application at distances of 0.3, 1.0, and 2.0 m from the trays in the low-tunnel field trial at the Experimental Lageado Farm (UNESP). The samples were stored at −20 °C until processing. The samples were prepared by macerating the leaves in liquid nitrogen, with subsequent weighing of 0.2 g of the material in 15 mL centrifuge tubes, to which 10 mL of methanol: distilled water (25:75 *v/v*) was added. The tubes were subjected to an ultrasound bath for 30 min, and after this period, the samples were centrifuged at 4000 rpm for 5 min at 20 °C. The supernatant was collected (1.5 mL), filtered, and placed in a vial for subsequent quantification of the dicamba.

The soybean line that showed the greatest injury damage within the tunnels was considered a criterion for evaluation, for both injury and the determination of dicamba. Once the application was performed, far from the experimental area, it was assumed that

all the symptoms observed were caused only by the volatilized dicamba fraction in the soil of the trays.

*2.3. Analytical Method for Quantification of Dicamba*

In the samples of the different experiments, dicamba was analyzed as described by Carbonari et al. [29], using a liquid chromatography–tandem mass spectrometry (LC–MS/MS) system composed of a high-performance liquid chromatograph (Prominence UFLC, Shimadzu, Kyoto, Japan) equipped with two LC-20AD pumps, an SIL-20AC autoinjector, a DGU-20A5 degasser, a CBM-20A controller system, and a CTO-20AC oven. The chromatograph was coupled to a Triple Quad 4500 mass spectrometer (Applied Biosystems, Foster City, CA, USA). Chromatographic analyses were performed with a C18 column (Phenomenex Gemini 5μ C18RP 110Å) using an injection volume of 20 μL, with 5 mM ammonium acetate (Avantor Performance Materials, Inc., Center Valley, PA, USA) in water and 5 mM ammonium acetate in methanol (Merck KGaA, Darmstadt, Germany). The flow rate used was 1.0 mL min$^{-1}$, and the ratio of the solvents was gradually increased from 80:20 (methanol/water) to 95:5 at 4 min and returned to the initial condition at 10 min. The total running time was 12 min, with retention in the chromatographic column of 5.68 min for dicamba. An electrospray ionization source (EIS) was used in a negative mode. Eight concentrations of the dicamba analytical standards with a certified purity level of 99.9% (Sigma Aldrich, St. Luis, MO, USA) were included in the calibration curve.

*2.4. Data Analysis*

For the study conducted in the laboratory, the data obtained after quantification of dicamba on the filters (ng) were converted to ng cm$^{-2}$ or percentage of the total deposition (sum of the total dicamba weight for each experimental unit). The data obtained in duplicate were submitted to analysis of variance independently, and the homogeneity test of residual variances (Fmax) was applied [33]. Confirming homogeneity between the experiments, the data were grouped in an experiment with six replications. From a new analysis of variance, the means were compared using the least significant difference (LSD) test, with a probability of 5%, using the statistical software Sisvar, version 5.6 (Sisvar®, Lavras, MG, Brazil) [34]. The confident interval ($p \leq 0.05$) for each mean were calculated.

The standard errors for each mean were calculated for the results of the quantification of dicamba in the field study. The injury data were fitted to a three-parameter sigmoidal nonlinear regression equation (Equation (1)):

$$y = a/(1 + \exp(-(x - x0)/b)) \tag{1}$$

where y is the response variable injury, x is the tray with dicamba, x0 is the distance that provides a 50% response of the variable, a is the difference between the maximum and minimum points of the curve, and b is the slope of the curve.

The injury ratio data as a function of the dicamba concentration in the plants were fitted to a two-parameter exponential nonlinear equation (Equation (2)):

$$y = a * (1 - \exp(-b * x)) \tag{2}$$

where y is the response variable (injury), x is the dicamba concentration in the plant tissue, a is the maximum value estimated for the response variable, and b is the slope of the curve.

## 3. Results

*3.1. Laboratory Results*

The DGA + VR treatment had the lowest level of volatility (85% reduction compared to that with the DGA alone). When compared with DGA alone, the association of DGA with glyphosate potassium salt did not result in increased volatilization. However, the combination of glyphosate ammonium and di-ammonium salts increased volatility (between 35 and 61%). When the VR was added to the solutions, there were significant reductions in

volatilization for DGA alone and for all mixtures with glyphosate when compared with the herbicides without VR. Among the associations, the treatment of DGA, glyphosate salt potassium, and VR showed a high reduction in volatility (77% compared to that with DGA alone). For the other glyphosate salts, there were significant reductions in volatility, but the volatility was always lower than that with DGA alone and that associated with the potassium salt (Figure 2).

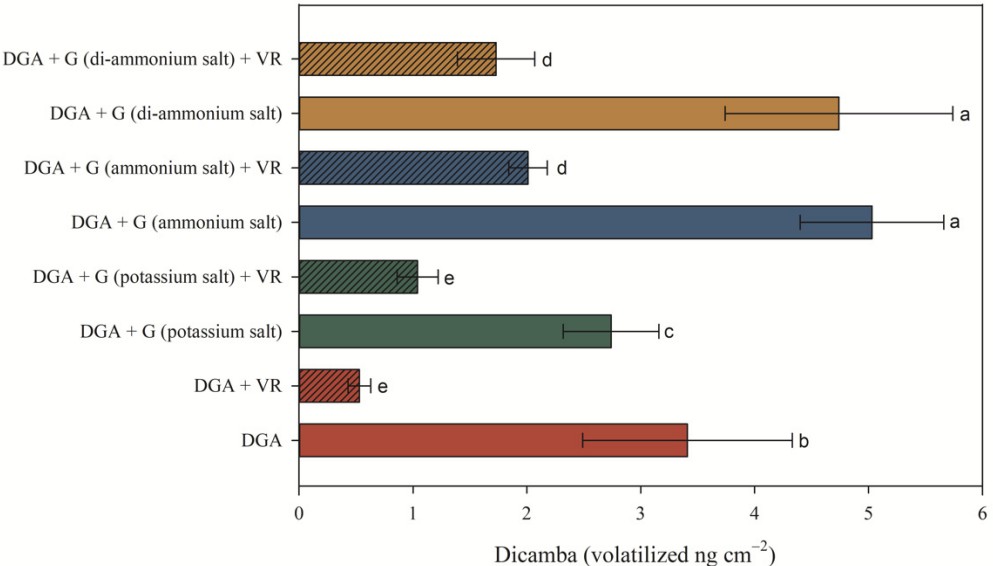

**Figure 2.** Volatilization of dicamba (ng cm$^{-2}$) alone and in combination with a volatility reducer and glyphosate formulations (potassium, ammonium, and di-ammonium salts). The error bars represent the confidence interval ($p \leq 0.05$). Averages followed by the same letter do not differ from each other according to LSD test ($p \leq 0.05$). VR: volatility reducer; DGA: dicamba diglycolamine salt; G: glyphosate.

The mixture of dicamba with ammonium salt and di-ammonium promoted a significant increase in the volatilization of dicamba. Although the presence of the VR minimized the volatilization losses even in these mixtures, the efficiency of the VR in the mixtures was lower than that observed for dicamba alone or dicamba mixed with glyphosate potassium salt (Figure 2).

*3.2. Low-Tunnel Field Trial Results*

Figure 1 shows thermal images of the internal and external areas of the plastic tunnels. The highest temperatures were observed in the center of the plots (area with bare soil, including trays), with maximum temperatures of 46.9 and 47.0 °C at 9:00 a.m. and 57.3 and 60.1 °C at 1:00 p.m., for the internal and external areas of the plastic tunnel, respectively. The average temperatures were 30.7 and 28.5 °C at 09:30 a.m. and 47.8 and 39.5 °C at 1:00 p.m. for the internal and external areas of the plastic tunnel, respectively. Thus, high temperatures were observed, a critical condition for dicamba volatilization.

The results of the dicamba concentrations in the soybean plants at different distances from the trays with dicamba showed that the application of the dicamba DGA salt alone caused the lowest dicamba contents in the soybean plants at the different distances from the tray, and for the application of the DGA + VR in the trays, no dicamba detection was observed in the soybean plants (Figure 3). The DGA + glyphosate salt potassium caused an intermediate level of herbicide present in the soybean plants, but the DGA + glyphosate salt potassium + VR was also very safe, with concentrations of the herbicide in the plants similar to those of dicamba alone (Figure 3). Despite the higher concentrations of dicamba in the soybean plants for these treatments, the presence of a VR promoted a slight reduction in volatilization. The results of injury in soybean plants indicate the same behaviors (Figure 4).

Soybean injuries caused by isolated DGA and DGA + glyphosate potassium salt were similar to each other and lower than the other mixtures (ammonium and di-ammonium salts). When VR was added to DGA and DGA + glyphosate potassium salt, the injury was even lower (Figure 4). Figure 5 shows the correlation between the percentages of injury in the plants and the concentrations of dicamba in the plants at the same distances. The maximum percentages of injury were approximately 40%, which is between 25 and 30 ng g$^{-1}$ of soybean dry mass.

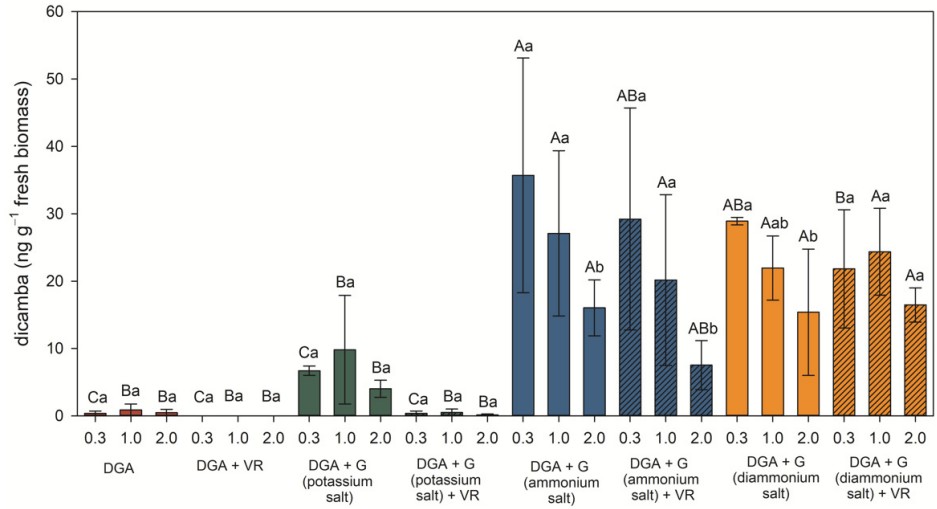

**Figure 3.** Dicamba concentrations in soybean plants at distances of 0.3, 1.0, and 2.0 m from the trays treated with dicamba in different associations and allocated within the low tunnels. The error bars represent the confidence interval ($p \leq 0.05$). Averages followed by the same upper-case letter do not differ from each other for the treatments (sprayed solution) according to LSD test ($p \leq 0.05$). Averages followed by the same lower-case letter do not differ from each other for the distances of trays according to the LSD test ($p \leq 0.05$). VR: volatility reducer; DGA: dicamba diglycolamine salt; G: glyphosate.

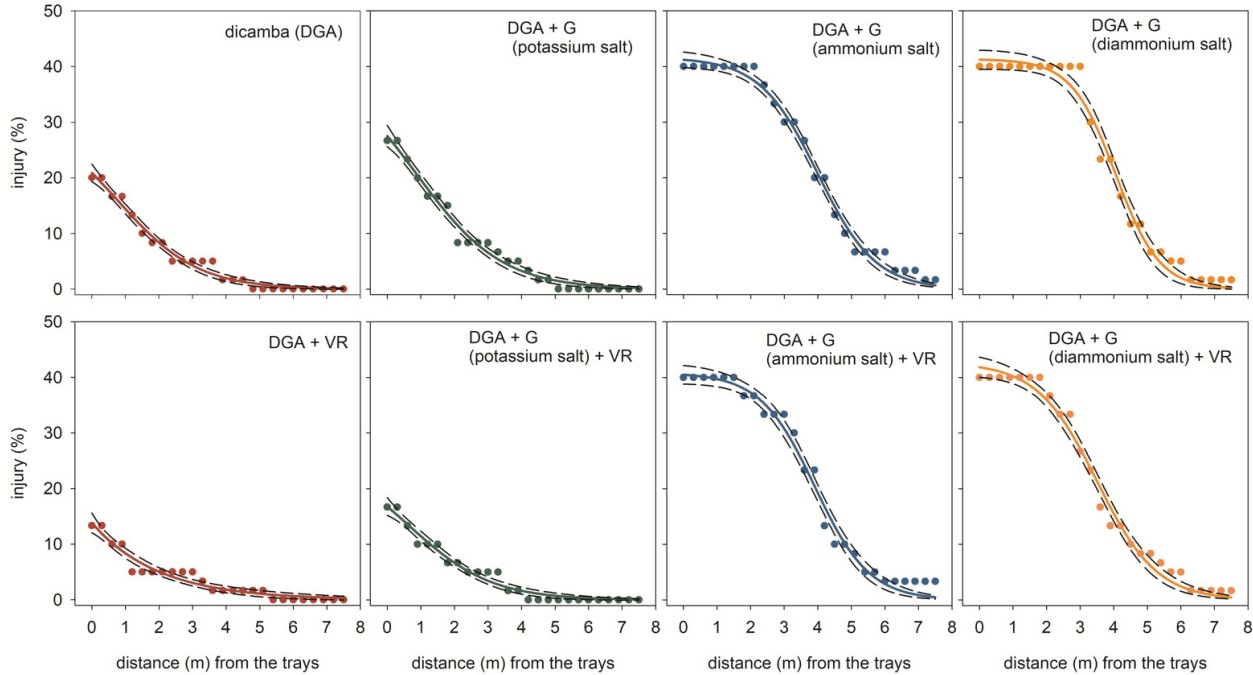

**Figure 4.** Soybean injury (%) for the different treatments in plants at different distances (m) from the trays treated with dicamba and allocated within the low tunnels. DGA: y = 31.70/(1 + exp (−(x − 0.78)/−1.19)),

$r^2$ 0.98; DGA+VR: y = 33.14/(1 + exp (−(x − 10.87)/−1.98)), $r^2$ 0.94; DGA + G potassium salt: y = 41.58/(1 + exp (−(x − 0.85)/−1.28)), $r^2$ 0.98; DGA+G potassium salt + VR: y = 26.77/(1 + exp (−(x − 0.64)/−1.23)), $r^2$ 0.99; DGA + G ammonium salt: y = 41.59/(1 + exp (−(x − 3.92)/−0.86)), $r^2$ 0.99; DGA + G ammonium salt + VR: y = 40.82/(1 + exp (−(x − 3.91)/−0.81)), $r^2$ 0.99; DGA + G di-ammonium salt: y = 41.31/(1 + exp (−(x − 4.05)/−0.65)), $r^2$ 0.98; DGA + G di-ammonium salt + VR: y = 42.66/(1 + exp (−(x − 3.50)/−0.89)), $r^2$ 0.98. The dashed lines indicate the confidence interval ($p \leq 0.05$). VR: volatility reducer; DGA: dicamba diglycolamine salt; G: glyphosate.

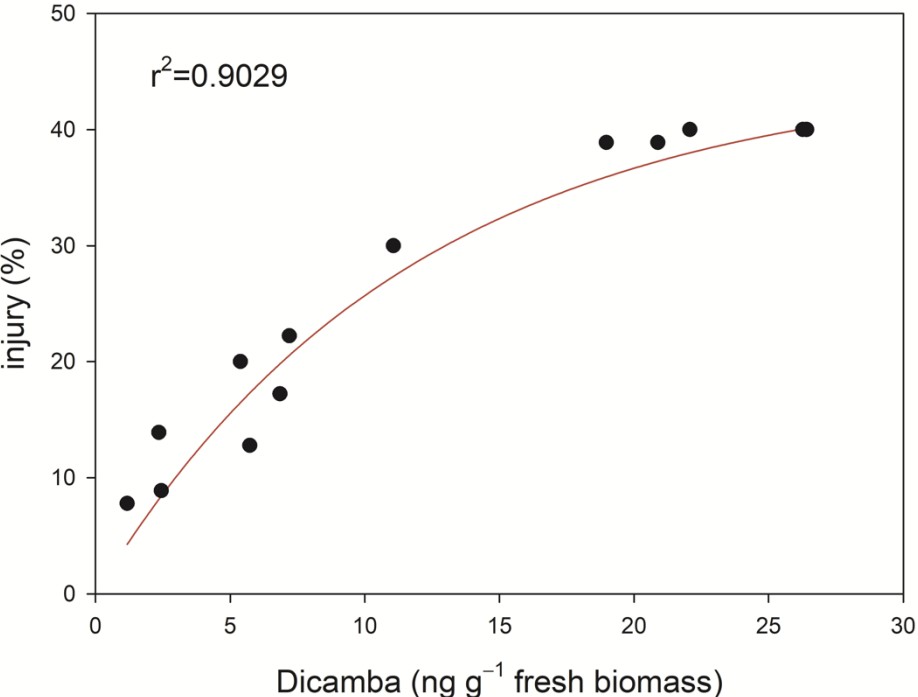

**Figure 5.** Regression fit between the mean soybean injury (%) and the dicamba concentrations in the plants at the related distances. Data points are the means of the replications, and the red line represents y = 44.85 ∗ (1 − exp (−0.085 ∗ x)).

## 4. Discussion

The temperature planned for the laboratory study, as well as the temperature found in the field study, were high, as reported. A high temperature and low humidity promote dicamba volatility [17–19], and in studies of dicamba volatility in low-tunnel field volatility experiments, Striegel et al. [35] observed the highest injuries to soybean at high air temperatures (maximum > 29 °C) and low wind speeds (mean 0.3–1.5 ms$^{-1}$) for 48 h after application of the treatments.

The pH of the mixtures observed in this study did not completely explain the differences in the volatilization level observed because similar values were found for some mixtures with important differences in volatility (Table 2 and Figure 2). The decrease in the pH of the solution due to the addition of glyphosate has been linked to an increase in dicamba volatilization [18]. Sharkey et al. [27] also found that the addition of glyphosate to dicamba DGA salt significantly decreased the pH value. However, when testing the BAPMA formulation of dicamba in combination with glyphosate, the pH value also decreased; however, the volatilization did not increase significantly, indicating that even with lower pH values, there may be no increases in volatilization.

The pKa of dicamba is 1.87 (Table 1), which indicates that at this pH, the numbers of nonionized and ionized molecules are equivalent. Therefore, the progressive increase in pH progressively increases the participation of ionized dicamba, which is more prone to volatilization. Therefore, the volatility of dicamba is dependent on pH. However, the use of

nonvolatile cations with strong binding to the dicamba anion has also been shown to be a highly effective option to reduce the volatility of this herbicide, ensuring a fundamental role for the dicamba formulation regarding volatilization potential. For example, DGA dicamba salt was less volatile than the DMA dicamba salt under field conditions [9,19], and in bioassays, it was demonstrated that the volatilization of dicamba applied in the form of DGA salt was reduced by 94% when compared to that of dicamba applied in the form of DMA salt [19]. The use of the VR also played a very important role, and the volatility reducer used in this study prevented dicamba ions from combining with hydrogen ions, significantly reducing the volatility potential [26].

Striegel et al. [35] evaluated the effect of a spray solution pH with different formulations of auxinic herbicides and tank additives on soybean injury caused by herbicide volatilization. Among all the treatment factors in the study, the inclusion of glyphosate was the one that most acidified the pH of the spraying solution; however, the addition of glyphosate did not influence soybean injury at different distances from the treated area in the low-tunnel field volatility experiment.

Oseland et al. [32] evaluated the effect of different soil pH values (ranging from 4.3 to 8.3) with a low-tunnel field trial and observed a large increase in soybean volatility and injury when dicamba was applied to the soil at a lower pH for all dicamba formulations tested (DGA, DGA + Vapor Grip® as a VR, BAPMA, and choline salt). In addition to the high temperatures recorded (Figure 1), the soil pH of the field experiment was 4.9, which most likely enhanced the volatility of dicamba, allowing us to evaluate the effects of the treatments under critical conditions for dicamba volatility. However, most of the dicamba applied to soybean areas in Brazil will be applied to plants and straw due to the predominance of the no-tillage system, a condition represented in the laboratory study.

The rate of dicamba volatilization may also depend on the type of surface to which it is applied. Carbonari et al. [29] found that even with the addition of potassium salt to dicamba liquid, volatilization did not increase for some surfaces, such as glass and wet soil. In the same study, for all treatments and surfaces, the addition of a VR significantly reduced volatilization, highlighting the importance of its use in dicamba applications.

The drift of dicamba vapor after spraying and deposition can be controlled by the dicamba formulation, which can be designed to reduce volatilization [27], or by the addition of additives to the tank mixture to reduce volatilization [29]. As glyphosate is also formulated as a salt, the inclusion of an additional amine or cation may have additional impacts on the volatilization of dicamba, in addition to the effects already reported for pH changes [27]. These authors [27] observed that the presence of free acid glyphosate in combination with dicamba and amines promoted high levels of volatility compared to those with free acid dicamba, even with an increase in the concentration of amine associated with dicamba.

Sall et al. [36] conducted twenty-three field experiments to provide an estimate of dicamba volatility after application of 0.56 or 1.12 kg ha$^{-1}$ and observed that the volatilization of all formulations and conditions tested ranged from $0.023 \pm 0.003\%$ to $0.302 \pm 0.045\%$ of the total dicamba applied, and the volatilization peaks occurred in the first 24 h after application. This information shows the importance of the first 24 h after application to the volatility of dicamba, and justifies the time adopted in this study.

Bish et al. [17] quantified dicamba in the air after its application and characterized the differences between the dicamba formulations of DGA salt + VR and BAPMA salt with and without glyphosate and under different meteorological conditions. The authors observed that the most volatile dicamba DGA salt and dicamba BAPMA salt were detected at similar levels over time when applied simultaneously. The highest concentrations for each formulation occurred from 0.5 to 8 h after application, and the concentrations of DGA + VR and BAPMA were 22.6 and 25.8 ng m$^{-3}$, respectively. Both formulations showed similarly rapid dissipation in the air, with dicamba concentrations decreasing from >20 ng m$^{-3}$ in 0.5 to 8 h after application (HAA) to <7 ng m$^{-3}$ between 8 and 16 HAA. The dicamba concentrations were <2 ng m$^{-3}$ and remained at this concentration until 72 HAA.

The levels of dicamba in the air were higher when glyphosate (potassium salt) was used. The not-observable adverse effect concentration of dicamba in the air for soybean plants is 138 ng m$^{-3}$ [37]. However, no reference values of dicamba concentrations in soybean plants that cause adverse effects were found in the literature. The high correlation between soybean injury levels and dicamba content in plant leaves demonstrated that the maximum percentages of injury were approximately 40%, which is between 25 and 30 ng g$^{-1}$ of soybean dry mass.

The low tunnels results were quite similar and allowed the same conclusions to be drawn from the laboratory studies and those presented above, which reinforces the quality and efficiency of the methods used. It is worth noting the importance of quantifying dicamba in plants in the field, in that this process provides a more precise differentiation between treatments in relation to the evaluation of injury only. In addition to these results (field and laboratory) having a good correlation, the results are strongly correlated with the information from the literature presented above [18]. The application conditions (four $\times$ the maximum dose of the package insert) and the confinement of the vapors with the plastic tunnels do not represent real conditions of herbicide use but allowed us to compare the risks of the different treatments.

## 5. Conclusions

The study methodologies discussed in this study are adequate for understanding the volatility of dicamba. The volatility of dicamba can be managed through the use of VRs and the correct formulation of products used in mixtures. Glyphosate potassium salt was shown to be the safest choice to combine with dicamba DGA salt, without increasing the volatility in relation to that with dicamba alone. The VR was efficient in reducing the volatility for dicamba alone and DGA in combination with all glyphosate salts. An association with lower volatility was observed for dicamba with glyphosate potassium salt and the VR.

**Author Contributions:** Conceptualization and funding acquisition, C.A.C. and E.D.V.; data curation, R.N.C., B.F.G. and N.C.B.; formal analysis, R.N.C., N.C.B., M.P. and C.A.C.; investigation, R.N.C., B.F.G., N.C.B., M.P., C.A.C. and E.D.V.; methodology development, C.A.C., R.N.C., B.F.G. and E.D.V.; supervision, C.A.C., E.D.V. and R.F.L.O.; writing original draft, C.A.C. and R.F.L.O.; review and editing, M.P. and H.B. All authors have read and agreed to the published version of the manuscript.

**Funding:** This research was funded by Bayer CropSciences.

**Institutional Review Board Statement:** Not applicable.

**Informed Consent Statement:** Not applicable.

**Data Availability Statement:** Not applicable.

**Acknowledgments:** The authors are grateful to Bayer CropSciences for funding and José Roberto Marques Silva for his support on the chemical analysis.

**Conflicts of Interest:** The authors declare no conflict of interest.

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
