# Peer review of "Volatilization of Dicamba Diglycolamine Salt in Combination with Glyphosate Formulations and Volatility Reducers in Brazil"

_agronomy, doi:10.3390/agronomy12051001_

Round 1

Reviewer 1 Report

Abstract - I would recommend adding Latin names to buva and caruru; in the abstract the M&M part is to long, should be shorter and present major results and conclusions.

line 43 - crops[1] - add space between words

line 49 - 2018[2,3] - add space between words

lines 51, 53, 55, 57, 64, 74, 76, 78, 104, 105, 111, 113, 142, 150, 162, 188, 201, 221, 264, 356, 371, 373,395,396 - delete fullstop or comma before citation in brackets, and add after brackets; e.g. registration.[4] change to registration [4].

line 59 - sediments[10] - add space between words

line 62 - volatile compound6 - is this reference 6 ?

line 134 – is this dose of g of a.i. per ha?

line 180 – 300 kg h-1 – do you mean 300 kg ha-1?

line 187 – Table 2, there is no Table 3

line 239 - Carbonari et al.[28] - add space between words

line 279 – Among the treatments

line 323, 324, 325  – please delete “his section may be divided by subheadings. It should provide a concise and precise description of the experimental results, their in-324 terpretation, as well as the experimental conclusions that can be drawn.”

line 409 - Bish et al.[16] - add space between words

Reference list - not all references are written according to the instructions, for example 1, 8 etc.

Author Response

Reviewer 1

We are grateful for your support to improve the manuscript. We try to accommodate all suggestions and corrections requested and we are available for new corrections if necessary.

Abstract - I would recommend adding Latin names to buva and caruru; in the abstract the M&M part is to long, should be shorter and present major results and conclusions.

You are right, we changed according recommended

line 43 - crops[1] - add space between words

This correction was made

line 49 - 2018[2,3] - add space between words

This correction was made

lines 51, 53, 55, 57, 64, 74, 76, 78, 104, 105, 111, 113, 142, 150, 162, 188, 201, 221, 264, 356, 371, 373,395,396 - delete fullstop or comma before citation in brackets, and add after brackets; e.g. registration.[4] change to registration [4].

This correction was made

line 59 - sediments[10] - add space between words

This correction was made

line 62 - volatile compound6 - is this reference 6?

yes, I changed format

line 134 – is this dose of g of a.i. per ha?

I added the information a.e. (acid equivalent)

line 180 – 300 kg h-1 – do you mean 300 kg ha-1?

yes, that was the amount used

line 187 – Table 2, there is no Table 3

you are right, I changed

line 239 - Carbonari et al.[28] - add space between words

This correction was made

line 279 – Among the treatments

This correction was made

line 323, 324, 325 – please delete “his section may be divided by subheadings. It should provide a concise and precise description of the experimental results, their in-324 terpretation, as well as the experimental conclusions that can be drawn.”

You are right, We deleted

line 409 - Bish et al.[16] - add space between words

This correction was made

Reference list - not all references are written according to the instructions, for example 1, 8 etc.

Correct, we have reviewed the entire references section

Reviewer 2 Report

“Volatilization of dicamba diglycolamine salt in combination with glyphosate formulations and volatility reducers in Brazil” is an interesting work with useful results for agricultural practice. But it must be carefully revised by the authors to correct some errors and improve the presentation of ideas. The results need to be better explained (based on statistical analyses).

Detailed comments:

Correct the references in the text throughout the manuscript. In many cases the punctuation is wrong and some spaces are missing. For example:

Line 43: “crops[1]”. Line 51: “registration.[4]”. Line 53: “leaves. [5,6].”

Consider including a section of abbreviations: BAPMA, DGA, DMA, VR, etc.

Lines 45-47: “In recent years, with the launch of technologies that allow the application of dicamba in postemergence soybean and cotton crops, there has been an intensification in the use of this herbicide in the United States of America”. Indicate which technologies or add a reference.

Line 62: “compound6”. Number 6 is the reference or is it a typographical error? Correct.

Table 1: Add the “Molecular weight” units (g mol-1).

Change “1.0 X 10-04” to “1.0 × 10-4”.

For “Koc” the acronym is used, but for “Octanol-water partition coefficient” it is not used. Be consistent.

Lines 124 and 192: Replace “vs.” by “VS”.

Lines 123-124 and 192: The model of the nozzles is indicated, but the trademark (TeeJet) is not indicated.

Line 125 and 194: "forward speed" is more suitable than "speed".

Line 126: “volume rate” or “spray volume” are more suitable than “spray consumption”.

Table 2 and Table 3: Why is the concentration indicated for VR and the dose for the herbicide? Or dose or concentration. Be consistent.

It would be interesting to indicate the concentration recommended by the manufacturer on the product label. Or indicate whether or not the recommended concentration has been used.

Table 2: Remove the “and” from the penultimate row.

Line 175: in “CaCl2” put the 2 in subscript.

Line 176: Replace “SB” by “Sb”.

Lines 174-177: Consider putting the physicochemical characteristics information in a table, and explain the meaning of the acronyms (MO, CEC, V%).

Change “mmolc” to “mmol” in the units.

Write correctly the units of amount of sand, clay and silt (g kg-1).

Lines 179-180: Correct the units “kg h-1” (ha).

Line 209: “Table 2.” is actually table 3.

Table 3: The note "f" does not appear in the footer.

Lines 218-219: “These evaluations were performed at 7, 14 and 21 days after application”. In Figure 4 the results by days are not shown. Explain this.

Line 219: The abbreviation "(DAA)" no longer appears in the text.

Lines 223-224: “at distances of 0.3, 1.0 and 2.0 meters from the trays” In the same row? from both rows to both sides of the tray? Explain it better.

Line 225: It is not necessary to repeat the coordinates again.

Line 248: Replace “mL minutes-1” by “mL min-1”.

Line 262: “least significant difference (LSD) t test”. The "t" is not necessary.

Line 265: The standard error is not a measure of dispersion, therefore it is advisable to indicate the value in parentheses next to the mean, instead of using ±. Mean (standard error).

But in the Figures presented in Results confidence interval bars are shown. Was the standard error or confidence interval calculated?

Subsection 3.1.: When the text says that there are significant or non-significant differences, indicate in parentheses the results of the statistical test that corroborate these statements.

Figure 2: indicate by means of letters (next to the error bars) the statistical differences between treatments.

Line 304: It is not indicated at what time the temperature of “57.3 and 60.1 °C” is reached.

Figure 3: On the x-axis, change the commas to points of the distances of “0.3”, “1.0” and “2.0” m.

Consider using a different colour of the bars for each treatment. For example, use the same colours (red, green, blue, and orange) but in a lighter shade when VR is included in the treatment.

And use the same colours for each treatment in all Figures, to make it easier for the reader to understand the results.

Subsection 3.2.: results of statistical analyses are not reported. Neither in the text nor in the Figure 3.

Briefly explain the results shown in Figure 4.

Figure 4: The coloured dot displayed next to the treatment name on each chart is unnecessary as only one treatment is shown on each chart. Consider removing it, since in some graphs it could be confused with data far from the curve.

Line 336: In "r2" the 2 must be written in superscript.

Figure 5: in the x-axis legend change “ng/g” to “ng g-1”.

Lines 323-325: Delete this: “his section may be divided by subheadings. It should provide a concise and precise description of the experimental results, their interpretation, as well as the experimental conclusions that can be drawn.”

Line 399: “These authors observed” which authors? Indicate.

Lines 417-418: “between 8 and 16 HAA between 24 and 48 HAA”. Correct.

Line 420: “(NOAEC)” this abbreviation does not appear in the text, so it is unnecessary to specify it.

Section “Author Contributions”: Some authors' initials are missing dots.

Line 450 and 452: Is it written “CropScience”, not “Cropsciences”. Check it.

Lines 522-523. Add the link of the website consulted in reference 37 (http://sitem.herts.ac.uk/aeru/ppdb/en/Reports/213.htm).

Author Response

Reviewer 2

“Volatilization of dicamba diglycolamine salt in combination with glyphosate formulations and volatility reducers in Brazil” is an interesting work with useful results for agricultural practice. But it must be carefully revised by the authors to correct some errors and improve the presentation of ideas. The results need to be better explained (based on statistical analyses).

We are grateful for your support to improve the manuscript. We try to accommodate all suggestions and corrections requested and we are available for new corrections if necessary.

Detailed comments:

Correct the references in the text throughout the manuscript. In many cases the punctuation is wrong and some spaces are missing. For example:

Line 43: “crops[1]”. Line 51: “registration.[4]”. Line 53: “leaves. [5,6].”

These corrections were made

Consider including a section of abbreviations: BAPMA, DGA, DMA, VR, etc.

We did not find in the instructions for authors how to insert an abbreviation section we tried to detail the abbreviations more carefully and/or remove them when not needed

Lines 45-47: “In recent years, with the launch of technologies that allow the application of dicamba in postemergence soybean and cotton crops, there has been an intensification in the use of this herbicide in the United States of America”. Indicate which technologies or add a reference.

We added the name of technology “In recent years, with the launch of Xtend® technology that allow the application of dicamba in postemergence…”

Line 62: “compound6”. Number 6 is the reference or is it a typographical error? Correct.

It is a reference. We corrected the format

Table 1: Add the “Molecular weight” units (g mol-1).

Change “1.0 X 10-04” to “1.0 × 10-4”.

For “Koc” the acronym is used, but for “Octanol-water partition coefficient” it is not used. Be consistent.

You are right, we changed the tree points

Lines 124 and 192: Replace “vs.” by “VS”.

This correction was made

Lines 123-124 and 192: The model of the nozzles is indicated, but the trademark (TeeJet) is not indicated. 

We added this information

Line 125 and 194: "forward speed" is more suitable than "speed".

This correction was made

Line 126: “volume rate” or “spray volume” are more suitable than “spray consumption”.

This correction was made

Table 2 and Table 3: Why is the concentration indicated for VR and the dose for the herbicide? Or dose or concentration. Be consistent.

It would be interesting to indicate the concentration recommended by the manufacturer on the product label. Or indicate whether or not the recommended concentration has been used.

This is the company's recommendation for the launch of this technology in Brazil, So it is possible to have flexibility for VR in different spray volume

Table 2: Remove the “and” from the penultimate row.

This correction was made

Line 175: in “CaCl2” put the 2 in subscript.

This correction was made

Line 176: Replace “SB” by “Sb”.

This correction was made

Lines 174-177: Consider putting the physicochemical characteristics information in a table, and explain the meaning of the acronyms (MO, CEC, V%).

Change “mmolc” to “mmol” in the units.

This correction was made

Write correctly the units of amount of sand, clay and silt (g kg-1).

This correction was made

Lines 179-180: Correct the units “kg h-1” (ha).

This correction was made

Line 209: “Table 2.” is actually table 3.

This correction was made

Table 3: The note "f" does not appear in the footer.

The footer was added. “fDose 4 times higher than regular dose for a critical condition for volatilization.”

Lines 218-219: “These evaluations were performed at 7, 14 and 21 days after application”. In Figure 4 the results by days are not shown. Explain this.

We did monitor at different times but chose to present the most critical moment of symptoms which occurred at 14 days after application. I corrected this sentence in M&M.

Line 219: The abbreviation "(DAA)" no longer appears in the text.

You are right, we removed this abbreviation

Lines 223-224: “at distances of 0.3, 1.0 and 2.0 meters from the trays” In the same row? from both rows to both sides of the tray? Explain it better.

Yes, the evaluations were performed in both rows (both sides of the tray. We changed added this information.

Line 225: It is not necessary to repeat the coordinates again.

No, I removed this information

Line 248: Replace “mL minutes-1” by “mL min-1”.

This correction was made

Line 262: “least significant difference (LSD) t test”. The "t" is not necessary.

This correction was made

Line 265: The standard error is not a measure of dispersion, therefore it is advisable to indicate the value in parentheses next to the mean, instead of using ±. Mean (standard error).

But in the Figures presented in Results confidence interval bars are shown. Was the standard error or confidence interval calculated?

Sorry, you are right. We used the confidence interval and the ± was not used because we chose to present in figures instead of tables. This correction was made

Subsection 3.1.: When the text says that there are significant or non-significant differences, indicate in parentheses the results of the statistical test that corroborate these statements.

We have added the letters of the mean comparison test in figures 2 and 3 so that it is clearer what the text says significant, non-significant or similar. Still, additional clarifications were made in some points of the text.

Figure 2: indicate by means of letters (next to the error bars) the statistical differences between treatments.

The letters from LSD test were included

Line 304: It is not indicated at what time the temperature of “57.3 and 60.1 °C” is reached.

This correction was made

Figure 3: On the x-axis, change the commas to points of the distances of “0.3”, “1.0” and “2.0” m.

This correction was made

Consider using a different colour of the bars for each treatment. For example, use the same colours (red, green, blue, and orange) but in a lighter shade when VR is included in the treatment.

And use the same colours for each treatment in all Figures, to make it easier for the reader to understand the results.

Your suggestion is very good. We use the same colors on all figures and highlight the VR treatments on the bar graphs (by hatching the bars). We tried to lighten the colors as recommended but it turned out too colorful, visually weird, and confusing.

Subsection 3.2.: results of statistical analyses are not reported. Neither in the text nor in the Figure 3.

The letters from LSD test were included

Briefly explain the results shown in Figure 4.

A brief explanation has been included

Figure 4: The coloured dot displayed next to the treatment name on each chart is unnecessary as only one treatment is shown on each chart. Consider removing it, since in some graphs it could be confused with data far from the curve.

This correction was made

Line 336: In "r2" the 2 must be written in superscript.

This correction was made

Figure 5: in the x-axis legend change “ng/g” to “ng g-1”.

This correction was made

Lines 323-325: Delete this: “his section may be divided by subheadings. It should provide a concise and precise description of the experimental results, their interpretation, as well as the experimental conclusions that can be drawn.”

This correction was made

Line 399: “These authors observed” which authors? Indicate.

The authors indication was made

Lines 417-418: “between 8 and 16 HAA between 24 and 48 HAA”. Correct.

We have made this correction “…decreasing from > 20 ng m-3 in 0.5 to 8 hours after application (HAA) to <7 ng m-3 between 8 and 16 HAA.”

Line 420: “(NOAEC)” this abbreviation does not appear in the text, so it is unnecessary to specify it.

This correction was made

Section “Author Contributions”: Some authors' initials are missing dots.

This correction was made

Line 450 and 452: Is it written “CropScience”, not “Cropsciences”. Check it.

This correction was made

Lines 522-523. Add the link of the website consulted in reference 37 (http://sitem.herts.ac.uk/aeru/ppdb/en/Reports/213.htm).

This correction was made
